# Relationship of Postoperative Pain and PONV after Minimally Invasive Surgery with the Serotonin Concentrations and Receptors’ Gene Polymorphisms

**DOI:** 10.3390/jpm11090833

**Published:** 2021-08-25

**Authors:** Natalia Ignaszak-Kaus, Antoni J. Duleba, Aleksandra Mrozikiewicz, Grażyna Kurzawińska, Agata Różycka, Jan Hauke, Michał Gaca, Leszek Pawelczyk, Paweł Piotr Jagodziński, Piotr Jędrzejczak

**Affiliations:** 1Department of Infertility and Reproductive Endocrinology, Poznan University of Medical Sciences, Polna 33, 60-535 Poznan, Poland; a.mrozikiewicz@gmail.com (A.M.); leszek.a.pawelczyk@gmail.com (L.P.); piotrjedrzejczak@gmail.com (P.J.); 2Department of Reproductive Medicine, University of California, La Jolla, San Diego, CA 92093, USA; antoni.duleba@yale.edu; 3Division of Perinatology and Women’s Diseases, Poznan University of Medical Sciences, 60-535 Poznan, Poland; gene@gpsk.am.poznan.pl; 4Laboratory of Molecular Biology, Division of Perinatology and Women’s Diseases, Poznan University of Medical Sciences, 61-701 Poznan, Poland; 5Department of Biochemistry and Molecular Biology, Poznań University of Medical Sciences, 60-781 Poznan, Poland; arozycka@ump.edu.pl (A.R.); pjagodzi@ump.edu.pl (P.P.J.); 6Faculty of Human Geography and Planning, Adam Mickiewicz University, 61-712 Poznan, Poland; jhauke@amu.edu.pl; 7Clinics of Anaesthesiology in Obstetrics and Gynecology, Poznan University of Medical Sciences, Polna 33, 60-535 Poznan, Poland; gacum@ump.edu.pl

**Keywords:** minimally invasive surgery, PONV, postoperative pain, infertility treatment, serotonin, polymorphisms of the serotonin receptors

## Abstract

(1) Background: there is a steady increase in the number of procedures performed via minimally invasive surgery, which have many benefits, but post-operative nausea and vomiting (PONV) and significant pain are still a common problem (2) Methods: 300 infertile women (18–40 years old) undergoing minimal invasive surgery. Interventions: laparoscopy and hysteroscopy performing, evaluation of postoperative symptoms, serotonin concentrations assessment, identify genetic polymorphisms. (3) Results: serotonin concentrations were significantly lower among women who required opioids (*p* = 0.006). The presence of the GG genotype in the rs6318 polymorphism of the 5HTR2C gene had a protective effect on PONV (OR = 0.503; C.I. = [0.300–0.841]; *p* = 0.008), when the GG variant of the rs11214763 polymorphism of the 5HTR3B gene, when the risk of PONV was 1.65-fold higher (OR = 1.652; C.I. = [1.003–2.723]; *p* = 0.048). Pain intensity was significantly higher among women with GG genotype of the rs6296 polymorphism of the 5HTR1B gene (OR = 1.660; C.I. = [1.052–2.622]; *p* = 0.029).; (4) Conclusions: the evaluation of serotonin concentration predicts requirement for opioid pain relief medication. The polymorphisms of the serotonin receptors affect the intensity of postoperative complaints.

## 1. Introduction

The past two decades have witnessed a steady increase in the number of procedures performed via minimally invasive surgery (MIS) including laparoscopies and hysteroscopies. Benefits of MIS include faster recovery, better cosmetic results and often improved outcomes. However, short-term sequelae include post-operative nausea and vomiting (PONV) and significant pain, affecting, respectively, approximately 30% [1], and up to 67–70% of patients [1,2,3]. The risk for PONV is higher in females than in males [4]. 

It has been proposed that a potential mechanism of action of sex hormones on mood, perception of pain and nausea related to effects on serotonin concentration [5,6]. Indeed, E2 (estradiol) and P4 (progesterone) reduce activity of monoamine oxidase type B, an enzyme metabolizing serotonin [5]. It has also been shown that the administration of estrogens increases tryptophan hydroxylase mRNA (a rate-limiting enzyme converting tryptophan to serotonin) [6]. Serotonin may also play a role in modulating vomiting since serotonin 5-HT_3_ receptor antagonists are effective and safe antiemetics [7,8]. In addition, serotonin participates in pain regulation [9,10,11]. The actions of serotonin may be affected by the polymorphisms of their receptors. Thus, for example, the CTT haplotype of one of the HTR3A polymorphisms has been shown to be associated with a greater risk of PONV, while the TAG haplotype is considered to be protective against PONV [12]. Several small studies have indicated that the polymorphism of serotonin receptors has an impact on pain sensation and response to analgesics [13,14].

The aims of the study were to analyze the incidence of PONV and postoperative pain following combined hysteroscopies and laparoscopies, and to determine whether serotonin play a role in these symptoms. 

## 2. Materials and Methods

### 2.1. Study Participants

The prospective study evaluated 300 infertile women undergoing laparoscopy combined with hysteroscopy. Mean age was 32.5 ± 5 years; mean BMI of patients: 23.3 ± 4.1 kg/m^2^. The subjects were recruited among the patients of the Clinic of Infertility and Reproductive Endocrinology, Poznan University of Medical Sciences, between January 2016 and January 2017. Women between 18 and 40 years old were deemed eligible and included in the study. Written informed consent was obtained from all participants. To minimize a potentially confounding effect of diet, and since tryptophan is a substrate for serotonin production [15], all subjects were instructed to follow a low-tryptophan diet for three days before surgery (excluding chocolate, bananas, eggs, seeds, nuts, dairy products and meat). The exclusion criteria were current hormone therapy, use of antidepressant drugs, use of opioids in the past, menstruation at the time of surgery, history of motion sickness, smoking, allergic reactions to pain relief and/or antiemetic medications, history of previous surgery and surgical risk greater than ASA II. We exclude patients with diabetes or obesity more than 35 BMI. 

#### Anesthesia and Analgesia

All patients were premedicated 1 h before surgery with midazolam (7.5 mg). Fentanyl, propofol and rocuronium were used to induce anesthesia. After preoxygenation by spontaneous ventilation, patients were anaesthetized and thereafter intubated. To minimize the risk of PONV, active ventilation before intubation was avoided. F_I_O_2_ (fraction of inspired oxygen) was maintained between 40–50% in all subjects. Fentanyl (every 20–30 min), rocuronium (every 30–40 min) and sevoflurane or desflurane were administered to maintain adequate anesthesia. Extubating was performed after respiratory muscle function returned to normal. Analgesic treatment was based on an analgesic ladder. When the severity of pain was >4 points on the NRS scale, an opioid was given and when the intensity of pain was lower, patients received a non-opioid drug (paracetamol, ketoprofen or diclofenac). If vomiting occurred, 4 mg of ondansetron was administered intravenously.

### 2.2. Laparoscopy and Hysteroscopy

Indications for the surgery were as follows: suspicion of endometriosis, polycystic ovary syndrome (PCOS), suspicion of tubal occlusion, ovarian cysts, uterine myomas or idiopathic infertility. Hysteroscopies was performed concomitantly with laparoscopies. At the time of laparoscopy, a Veress needle was placed via a small periumbilical incision followed by an insufflation of carbon dioxide to attain intraperitoneal gas pressure of 12–14 mmHg pressure (2–3 L of CO_2_). Subsequently, the laparoscope and an additional 2–3 trocars were inserted, as required. Each patient underwent chromopertubation using methylene blue to assess the patency of the fallopian tubes. Mean operative time was 47.55 ± 29.5 min.

### 2.3. Evaluation of Postoperative Symptoms

All subjects were given a questionnaire and instructions on the day of the admission to hospital, one day before the procedure. Detailed instructions were provided for patients to self-assess postoperative pain, nausea and vomiting. Pain was evaluated using the Numerical Rating Scale [16], whereas PONV intensity was assessed on the scale from 0 to 2 (0—No complaints, 1—Nausea, 2—Nausea and vomiting). Patient requirement for antiemetic and pain relief drugs postoperatively was analyzed. Data were obtained from medical history and medical order sheets. In the presence of vomiting or significant pain, patients were closely monitored and given additional dose (s) of medications, in accordance with the WHO analgesic ladder. Due to the possible effect of the anesthetic agents administered during the surgery, and anesthesiologist-prescribed opioid pain relief, the study evaluated PONV and pain starting at 6 h after the procedure, when the patients were fully awake, followed by subsequent evaluations at 12 and at 24 h post-surgery.

### 2.4. Laboratory Tests

Fasting blood samples were collected at 6 a.m. on the day of the surgery. An enzyme-linked immunosorbent assay (ELISA) was used to measure serotonin concentrations. Genomic DNA was extracted from peripheral blood leukocytes using the QIAGEN System (QIAmp DNA Blood Midi Kit; QIAGEN, Hilden, Germany), according to the manufacturer’s protocol, to identify genetic polymorphisms (Table 1 presents details of the testing of polymorphisms). PCR-restriction fragment length (PCR-RFLP) combination analysis was used to identify the gene polymorphisms. After the PCR process, the obtained fragments of DNA underwent electrophoresis.

### 2.5. Statistical Analysis

Statistical analysis was performed using Statistica 13.1 program. *p*-Values < 0.05 were considered significant. Continuous variables are presented as means ± SD. Means of normally distributed variables were compared by Student’s *t*-test, or in the absence of homogeneity of variance, by Welch’s *t*-test, In the absence of normal distribution, the U Mann-Whitney test was applied. Distribution of categorical variables was evaluated by the Chi-Square Test or the Fisher Exact test, as appropriate. Aggregate scores for PONV and pain were calculated as sum of scores at 6, 12 and 24 h after surgery. Standard aggregate scores were calculated as the positive or negative fractional number of standard deviations by which the value of an aggregate score was above the mean value. Patients whose aggregate scores were negative experienced less than average pain or nausea and were assigned to the ‘low intensity pain’ or ‘low intensity PONV’ group, whereas positive aggregate score corresponded to more pain or nausea (‘high intensity pain’ or ‘high PONV’ group). Standardized data aggregation allowed measuring the intensity of complaints in one individual in comparison to the rest of the group. To determine the relationship between polymorphisms of analyzed genes and the prevalence of PONV and postoperative pain, Fisher’s exact test and the Chi-Square Test have been used. Odds ratio (OR) and 95% confidence intervals were calculated. The Hardy-Weinberg distribution was evaluated using an online calculator (http://www.oege.org/software/hwe-mr-calc.shtml (accessed on 10 January 2019)).

## 3. Results

### 3.1. PONV and Serotonin

High-intensity PONV was observed in 121 women (40.3% of all study participants). No relationship between serotonin concentration and PONV intensity was found. Mean serotonin concentrations in the low-intensity and high-intensity PONV groups were 183.5 ± 138.4 ng/mL and 175.65 ± 132.1 ng/mL, respectively (*p* = 0.618).

### 3.2. Pain and Serotonin

High intensity pain was reported by 144 subjects (48%). Despite the absence of a significant relationship between self-reported pain intensity (aggregate score of subjective assessment of the patient within a given group) and blood serotonin concentrations (*p* = 0.18), a statistically significant relationship was observed between the requirement for opioid pain relief medication (morphine or tramadol) and serotonin concentrations. Serotonin concentration was significantly lower among women who required opioid administration (mean 160.35 ± 104.0 ng/mL) as compared to levels of serotonin (mean 216.15 ± 182.1 ng/mL) in women who did not require opioids (*p* = 0.001). The relationship between serotonin concentrations and opioid requirement is presented in Figure 1. Since opioids exert an emetogenic effect, we tested for the presence of such a correlation among our study population. No relationship was found between the administration of pain relief medication (morphine and tramadol) and PONV intensity (*p* = 0.12). 

### 3.3. PONV, Pain and Receptors’ Gene Polymorphisms

The distribution of all investigated genotypes was consistent with the Hardy-Weinberg equilibrium (no statistically significant differences in the distribution between the observed and the expected frequencies for individual genotypes were found). The evaluation of the genetic polymorphism of the serotonin receptors (rs6318 polymorphism of the 5HTR2C gene) revealed that the CC genotype was associated with lowered odds of development of PONV (OR= 0.50; C.I.= [0.30–0.84]; *p* = 0.008). The presence of the GG genotype of the rs1121476 polymorphism of the 5HTR3B gene was associated with a 1.65-fold higher risk for PONV (OR = 1.65; C.I.= [1.003–2.72]; *p*= 0.048), whereas the presence of the GG genotype of the rs6296 polymorphism of the 5HTR1B gene was associated with an increased risk for high intensity postoperative pain (OR= 1.66; C.I.= [1.05–2.62]; *p*= 0.029). The distribution of alleles and genotypes of the analyzed polymorphisms and the odds ratio score for (PONV and post-operative pain, respectively, is presented in Table 2 and Table 3. 

## 4. Discussion

The present study provides several relevant findings regarding post-operative symptoms in women undergoing laparoscopy and hysteroscopy: (i) postoperative pain and PONV are still an important problem; (ii) serotonin concentration was significantly lower among women who required opioids six or more hours after surgery; and (iii) polymorphism of several genes (encoding serotonin receptors) is associated with greater (5HT3B) and lower (5HTR2C) risk for PONV and greater risk for post-operative pain (5HTR1B). 

Pain after laparoscopy may be at the incision site—‘surgical wound pain’ and ‘non-surgical wound pain’—or be located in the supraclavicular area (referred pain) [23]. Carbon dioxide, which is necessary to create the pneumoperitoneum, is the main cause behind postoperative shoulder pain because it affects the diaphragm and stimulates the phrenic nerve, resulting in pain in the fourth neck dermatome [24,25]. The dual nature of postoperative pain, typical for endoscopic procedures, may be the reason why we did not observe any relationships between serotonin concentration and the intensity of abdominal complaints, but found a link between serotonin concentration and requirement for opioid drugs. Patients who did not require opioid pain relief medication had higher levels of serotonin (*p* = 0.006). Opioid analgesics act predominantly at the injury site (surgical pain), which is the result of nociceptor stimulation [26,27], rather than non-surgical pain. In our study, we demonstrated a protective effect of the CC variant of the rs6318 polymorphism of the 5HTR2C gene as its carriers were less likely to experience nausea and vomiting. To our knowledge, the rs6318 polymorphism of the 5HTR2C gene has not been previously investigated with regard to postoperative nausea and vomiting.

There are plenty of different factors which influence levels of serotonin in the blood, ex.: diet rich in tryptophan [15], diabetes type 2 [28] obesity [29], previous surgery- these dependences were mentioned in the Martin AM et al. study [30]. In our study we excluded the patients with diabetes and obesity more than 35 BMI; also, all subjects were operated for the first time and followed a special diet before planned surgery.

To assess PONV we used 0/1/2 scale which was easy to understand ad explain for patients. Nausea was defined for patients as feeling sickness in the stomach and sensation of unease and discomfort, often perceived as an urge to vomit. Retching and vomiting were not distinguished as in other studies. In the study of McKenzie and al. [8] authors count the number of vomiting episodes and used 0–10 scale (0—No vomiting, 10—Nausea as bad it can possible be). In the study of McKeen and al., the authors used another distinction- nausea was assessed as: none, mild, moderate or severe; vomiting was any emetic episode or retching [31]. 

In the study of McKeen DM et al., the authors assessed the role of intraoperative supplemental oxygenation in prevention of PONV after gynecological laparoscopy [31]. They demonstrated no difference between both groups (80% vs. 30% oxygen). In our study F_I_O_2_ (fraction of inspired oxygen) was maintained between 40–50% in all subjects so this factor could not influence on difference in PONV. 

We also found that the carriers of the GG genotype of the rs11211476 polymorphism (5HTR3B gene) had a statistically significantly higher risk for PONV. Varying intensity of nausea in patients with different variants of this polymorphism indicates that despite being located in the intron of the 5HTR3B gene, it may play a role in gene splicing and affect its expression [32]. 

In a study by Lehmann et al., pregnant carriers of the CC variant of the rs1176744 gene polymorphism of the HTR3B receptor gene required higher doses of antiemetic drugs, while the AG variant in case of the rs3782025 polymorphism was associated with lower susceptibility to vomiting [33]. In another multicenter study conducted among 1600 oncology patients, numerous correlations between genetic variants of the HTR3B serotonin receptor and the intensity of opioid therapy-induced nausea and vomiting were demonstrated [22]; notably, this study evaluated chemotherapy-induced nausea and vomiting (CINV) and not postoperative symptoms. Some genetic studies have demonstrated the role of single nucleotide polymorphisms different than 5-HT receptors’ genes in nausea and antiemetic drugs metabolism what was widely described in Candiotti K et al. [34]. Differences in pain sensitivity may also depend on serotonin receptor polymorphisms. Interestingly, in one study, carriers of the AC genotype of the rs1176744 polymorphism (HTR3B receptor) experienced higher-intensity pain, but this correlation was observed only in female carriers [14]. In contrast, Klepstad P et al. found no relationship between any genotype (including genetic variants of HTR3A as well as HTR3B and different sensitivity or resistance to opioid pain relief medications among almost 3000 oncology patients [35]. 

In our study, we demonstrated that the carrier state of the GG genotype of the rs6296 polymorphism of the 5HTR1B gene was linked with a 1.66-fold higher risk for postoperative pain. Carriers of the GC and CC variants reported significantly lower-intensity complaints (*p* < 0.05). To date, no clinical significance of this polymorphism has been documented [36]. It is a synonymous mutation resulting in no change in the amino acid of the encoded protein. It may, however, affect the rate of protein synthesis and folding-which determines its structure and hence may affect the phenotype [37]. Differences in pain intensity detected among carriers of variants of the rs6296 polymorphism might be examples of such a process. Further studies are needed to clarify the relationship of genetic variants of investigated receptors and postoperative pain/PONV intensity. Such studies may contribute to the development of new drugs. Further studies with more subjects including control group are needed to clarify the pain and PONV pathogenesis and probably will allow for more precise controlling postoperative complaints.

## 5. Conclusions

Pain as well as nausea and vomiting after gynecologic laparoscopy and hysteroscopy are common and constitute a significant clinical problem. Serotonin evaluation may be a useful predictor of the requirements for opioid pain relief drugs. Polymorphism rs6318 of the 5HTR2C gene and rs11214763 of the 5HTR3B gene have an effect on the severity of postoperative nausea and vomiting, while the polymorphism of rs6296 of the 5HTR1B gene plays a role in the intensity of postoperative pain.

## Figures and Tables

**Figure 1 jpm-11-00833-f001:**
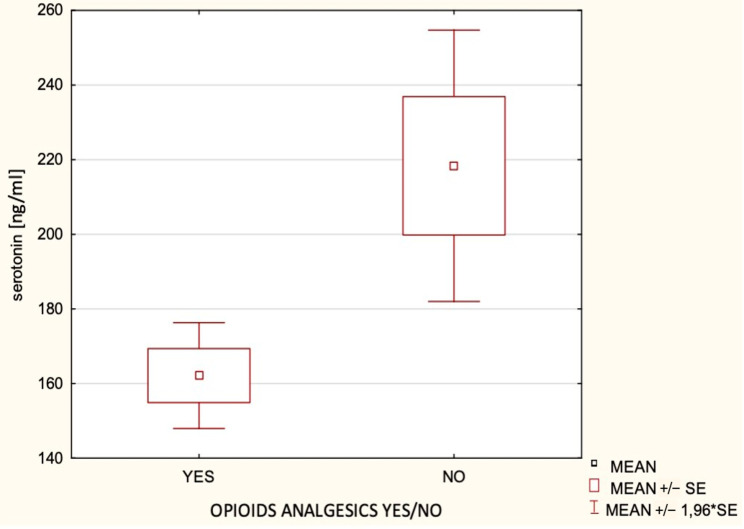
The relationship between serotonin concentrations and opioid requirement.

**Table 1 jpm-11-00833-t001:** Characteristics of the investigated polymorphisms.

Gene	SNP Number	Localization	Functional	Source
5HTR1B	rs6296	Val287Val	NO	[17]
5HTR2C	rs6318	Cys23Ser	YES	[18]
5HTR3B	rs11214763	Intron	NO	[19]
5HTR3B	rs1672717	Intron	YES	[20]
5HTR3B	rs1176744	Tyr129Ser	YES	[21]
5HTR3B	rs3782025	Intron	YES	[22]

5HTR—5-hydroxytryptamin receptor; SNP—single-nucleotide polymorphism.

**Table 2 jpm-11-00833-t002:** Distribution of alleles and genotypes of candidate gene polymorphisms in the etiology of postoperative nausea and vomiting (PONV) in a group of women operated on by hysteroscopy and gynecological laparoscopy due to infertility, depending on the severity of the ailments.

Polymorphism	Number (*n*)	Genotype and Allele Frequency					*p*	Genotype Comparative Analysis OR (95% CI); *p*
		**GG**	**GC**	**CC**	**G**	**C**		
*5HTR2C* (rs 6318)	PONV (300 patients)	202 (0.67)	90 (0.30)	8 (0.03)	494 (0.82)	106 (0.18)		comparative assessment of odds ratio for nausea in women, depending on symptom intensity, with GG and GC genotype vs. CC OR = 0.503; C.I. = [0.300–0.841]; *p* = 0.008
	low-intensity	110	64	5	284	74		
	high-intensity	92	26	3	210	32	*p* = 0.019	
		**GG**	**GC**	**CC**	**G**	**C**		
*5HTR1B* (rs 6296)	PONV (300 patients)	153 (0.51)	112 (0.37)	35 (0.12)	418 (0.70)	182 (0.30)		comparative assessment of odds ratio for nausea in women, depending on symptom intensity, with GG and GC genotype vs. CC OR = 0.556; C.I. = [0.250–1.239]; *p* = 0.147
	low-intensity	89	65	25	243	115		
	high-intensity	64	47	10	175	67	*p* = 0.246	
		**TT**	**TG**	**GG**	**T**	**G**		
*5HTR3B* (rs 1176744)	PONV (300 patients)	142 (0.47)	134 (0.45)	24 (0.08)	418 (0.70)	182 (0.30)		comparative assessment of odds ratio for nausea in women, depending on symptom intensity, with CC and CT genotype vs. TT OR = 1.164; C.I. = [0.483–2.805]; *p* = 0.735
	low-intensity	88	77	14	253	105		
	high-intensity	54	57	10	165	77	*p* = 0.515	
		**GG**	**GA**	**AA**	**G**	**A**		
*5HTR3B* (rs 11214763)	PONV (300 patients)	210 (0.70)	85 (0.28)	5 (0.02)	505 (0.84)	95 (0.16)		comparative assessment of odds ratio for nausea in women, depending on symptom intensity, with GG genotype vs. GA and AA OR = 1.652; C.I. = [1.003–2.723]; *p* = 0.048
	low-intensity	133	44	2	310	48		
	high-intensity	77	41	3	195	47	*p* = 0.048	
		**CC**	**CT**	**TT**	**C**	**T**		
*5HTR3B* (rs 3782025)	PONV (300 patients)	77 (0.25)	152 (0.51)	71 (0.24)	306 (0.51)	294 (0.49)		comparative assessment of odds ratio for nausea in women, depending on symptom intensity, with CC and CT genotype vs. TT OR = 0.966; C.I. = [0.500–1.867]; *p* = 0.919
	low-intensity	46	90	43	182	176		
	high-intensity	31	62	28	124	118	*p* = 0.923	
		**TT**	**TC**	**CC**	**T**	**C**		
*5HTR3B* (rs 1672717)	PONV (300 patients)	99 (0.33)	157 (0.52)	44 (0.15)	355 (0.59)	245 (0.41)		comparative assessment of odds ratio for nausea in women, depending on symptom intensity, with TT and TC genotype vs. CC OR = 1.031; C.I. = [0.503–2.114]; *p* = 0.933
	low-intensity	57	97	25	211	147		
	high-intensity	42	60	19	144	98	*p* = 0.890	

**Table 3 jpm-11-00833-t003:** Allele and genotype frequency of polymorphisms of encoding candidate genes in the etiology of postoperative pain in women undergoing hysteroscopy and gynecologic laparoscopy due to infertility, depending on symptom intensity.

Polymorphism	Number (*n*)	Genotype and Allele Frequency					*p*	Genotype Comparative Analysis OR (95% CI); *p*
		**GG**	**GC**	**CC**	**G**	**C**		
*5HTR2C* (rs 6318)	PAIN (300 patients)	202 (0.67)	90 (0.30)	8 (0.03)	494 (0.82)	106 (0.18)		comparative analysis of odds ratio for pain in women, depending on symptom intensity, with GG and GC genotype vs. CC OR = 1.082; C.I. = [0.263–4.448]; *p* = 0.912
	low-intensity	105	47	4	257	55		
	high-intensity	97	43	4	237	51	*p* = 0.979	
		**GG**	**GC**	**CC**	**G**	**C**		
*5HTR1B* (rs 6296)	PAIN (300 patients)	153 (0.51)	112 (0.37)	35 (0.12)	418 (0.70)	182 (0.30)		comparative analysis of odds ratio for pain in women, depending on symptom intensity, with GG genotype vs. GC and CC OR = 1.660; C.I. = [1.052–2.622]; *p* = 0.029
	low-intensity	89	50	17	228	84		
	high-intensity	64	62	18	190	98	*p* = 0.058	
		**TT**	**TG**	**GG**	**T**	**G**		
*5HTR3B* (rs 1176744)	PAIN (300 patients)	142 (0.47)	134 (0.45)	24 (0.08)	418 (0.70)	182 (0.30)		comparative analysis of odds ratio for pain in women, depending on symptom intensity, with TT and TG genotype vs. GG OR = 1.185; C.I. = [0.498–2.815]; *p* = 0.701
	low-intensity	77	67	12	221	91		
	high-intensity	65	67	12	197	91	*p* = 0.517	
		**GG**	**GA**	**AA**	**G**	**A**		
*5HTR3B* (rs 11214763)	PAIN (300 patients)	210 (0.70)	85 (0.28)	5 (0.02)	505 (0.84)	95 (0.16)		comparative analysis of odds ratio for pain in women, depending on symptom intensity, with GG and GA genotype vs. AA OR = 1.588; C.I. = [0.260–9.700]; *p* = 0.613
	low-intensity	108	46	2	262	50		
	high-intensity	102	39	3	347	45	*p* = 0.893	
		**CC**	**CT**	**TT**	**C**	**T**		
*5HTR3B* (rs 3782025)	PAIN (300 patients)	77 (0.25)	152 (0.51)	71 (0.24)	306 (0.51)	294 (0.49)		comparative analysis of odds ratio for pain in women, depending on symptom intensity, with CC and CT genotype vs. TT OR= 0.989; C.I. = [0.518–1.888]; *p* = 0.973
	low-intensity	41	77	38	159	153		
	high-intensity	36	75	33	147	141	*p* = 0.984	
		**TT**	**TC**	**CC**	**T**	**C**		
*5HTR3B* (rs 1672717)	PAIN (300 patients)	99 (0.33)	157 (0.52)	44 (0.15)	355 (0.59)	245 (0.41)		comparative analysis of odds ratio for pain in women, depending on symptom intensity, with TT and TC genotype vs. CC OR = 1.062; C.I. = [0.522–2.162]; *p* = 0.867
	low-intensity	51	83	22	185	127		
	high-intensity	48	74	22	170	118	*p* = 0.947	

## Data Availability

The data presented in this study are available on request from the corresponding author.

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
