# Peer review of "Relationship of Postoperative Pain and PONV after Minimally Invasive Surgery with the Serotonin Concentrations and Receptors’ Gene Polymorphisms"

_jpm, 2021, doi:10.3390/jpm11090833_

Round 1
Reviewer 1 Report
The manuscript “Relationship of postoperative pain and PONV after minimally invasive surgery with the serotonin concentrations and receptors’ gene polymorphisms” is well prepared and I think it provides new and valuable information about the role played by serotonin after laparoscopy and hysteroscopy. The aims of the study were to analyze the incidence of pain and post-operative nausea and vomiting after laparoscopy and hysteroscopy, and to determine whether serotonin play a role in these symptoms in order to use serotonin evaluation as a predictor of the requirements for opioid pain relief drugs. The experiments are well designed and the results clearly presented.
However, from my point of view there are some corrections that need to be made. First, there is no author affiliation on “Institute of Socio-Economic Geography and Spatial Management, Adam Mickiewicz University, Poznan, Poland” (5). Second, there is some repeated information. For example in first page, row 46: “of patients”; in page 10, rows 20-26: “To our knowledge, the rs6318 polymorphism of the 5HTR2C gene has not been previously investigated with regard to postoperative nausea and vomiting.”
Author Response
Please see the attachment
Thank you in advance

Reviewer 2 Report
Authors present a retrospective study on 300 infertile women (18-40 25
years old) undergoing minimal invasive laparoscopy and hysteroscopy and evaluate postoperative symptoms, serotonin concentrations with identification of gennetic polymorphisms. Serotonin concentrations were found to be significantly lower among women who required opioids and the presence of the GG genotype in the rs6318 polymorphism of the 5HTR2C gene had a protective effect on postoperative nausea and vomiting (PONV). Pain intensity was significantly higher amongwomen with GG genotype of the rs6296 polymorphism of the 5HTR1B gene. Authors conclude that the evaluation of serotonin concentration predicts requirement for opioid pain relief medication and that the polymorphisms of the serotonin receptors affect the intensity of postoperative complaints. Introduction provides sufficient information and Materials and Methods part was fairly written. Please state in the study design if this is prospective or retrospective evaluation; it looks more like a retrospective evaluation of prospectively gathered data. Evaluation of PONV is too simplified with 0/1/2 scale which makes the interpretation of the results complicated.Serotonin concentration was singificantly lower among women who required opioid administration compared to levels of serotonin in women who did not require opioids. There is a potential bias to this statistical signficance due to the fact that the women who needed opiod administration could have been patients with concomitant diseases, previous surgeries, previous use of opiods, especially if previously operated on spine - please comment on this; if possible, provide these general information for both groups (opiod vs. non-opiod).
There are numerous different factors which influence levels of serotonin in the blood. I suggest expanding the discussion for this possible confounding factors and including following publications:
Martin AM, Young RL, Leong L, Rogers GB, Spencer NJ, Jessup CF, Keating DJ. The Diverse Metabolic Roles of Peripheral Serotonin. Endocrinology. 2017 May 1;158(5):1049-1063. doi: 10.1210/en.2016-1839. PMID: 28323941.
McKeen DM, Arellano R, O'Connell C. Supplemental oxygen does not prevent postoperative nausea and vomiting after gynecological laparoscopy. Can J Anaesth. 2009 Sep;56(9):651-7. doi: 10.1007/s12630-009-9136-4. Epub 2009 Jul 29. PMID: 19639376.
Furthermore, there are several publications which investigate this matter which should be thoroughly discussed, for example:
Candiotti K, Shrestha C, Silva Ceschim MR. Is there a place for genetics in the management of PONV? Best Pract Res Clin Anaesthesiol. 2020 Dec;34(4):713-720. doi: 10.1016/j.bpa.2020.05.002. Epub 2020 May 13. PMID: 33288121.
Please include a statement on possible implications of your research and future directions. Include low number of patients, absence of control group and retrospective character (if so) of the study as clear limitations.
Author Response

(The authors gave the same response as above.)

Round 2
Reviewer 2 Report
The authors have explained all the requests